# Enhancing Predictability Assessment: An Overview and Analysis of Predictability Measures for Time Series and Network Links

**DOI:** 10.3390/e25111542

**Published:** 2023-11-15

**Authors:** Alexandra Bezbochina, Elizaveta Stavinova, Anton Kovantsev, Petr Chunaev

**Affiliations:** National Center for Cognitive Research, ITMO University, 16 Birzhevaya Lane, Saint Petersburg 199034, Russia; alexandra.bezbochina@yandex.ru (A.B.); ankovantcev@itmo.ru (A.K.); chunaev@itmo.ru (P.C.)

**Keywords:** predictability measures, time series analysis, intrinsic predictability, realized predictability

## Abstract

Driven by the variety of available measures intended to estimate predictability of diverse objects such as time series and network links, this paper presents a comprehensive overview of the existing literature in this domain. Our overview delves into predictability from two distinct perspectives: the *intrinsic* predictability, which represents a data property independent of the chosen forecasting model and serves as the highest achievable forecasting quality level, and the *realized* predictability, which represents a chosen quality metric for a specific pair of data and model. The reviewed measures are used to assess predictability across different objects, starting from time series (univariate, multivariate, and categorical) to network links. Through experiments, we establish a noticeable relationship between measures of realized and intrinsic predictability in both generated and real-world time series data (with the correlation coefficient being statistically significant at a 5% significance level). The discovered correlation in this research holds significant value for tasks related to evaluating time series complexity and their potential to be accurately predicted.

## 1. Introduction

Nowadays, the task of forecasting is a crucial topic with wide-range applications and significance in various fields. In fact, forecasting provides valuable insights into the future, helping individuals, organizations, and societies plan and adapt in a dynamic and uncertain world. Time series [1,2] and network link [3,4] are among the most significant and extensively investigated objects that have been the focus of forecasting efforts. The range of developed forecasting models is wide, starting from regression models [5] to neural networks [6] for time series forecasting, and from well-established classification methods [7] to random walks [8] for network links prediction.

Despite the extensive range of developed forecasting methods, researchers rarely address questions concerning the overall predictability of an object and its upper bound. Nevertheless, there are studies that delve into this matter, and the predictability measures they propose encompass a wide range, reflecting diverse viewpoints from various scientific domains, including dynamical systems [9] and information theory [10]. Our objective in this paper is to clarify the differentiation between prediction accuracy and predictability. We offer a comprehensive survey within the domain of predictability and the measures employed to evaluate it.

The concept of predictability is usually regarded from two sides [11]: with respect to a chosen forecasting model (*realized* predictability) and as a data property not depending on a certain model (*intrinsic* predictability). A significant part of this research is dedicated to the analysis of the correlation between realized and intrinsic predictability measures. Despite the extensive body of work on estimating object (such as time series [11] or network links [12]) predictability, this topic needs further exploration, particularly in regards to investigating the relationship between forecasting errors and intrinsic predictability measures. Gaining a comprehensive understanding of the relationship between forecast accuracy and time series complexity, which can be gauged using established intrinsic predictability measures, holds potential for assessing the achievable level of forecast accuracy (up to the forecasting moment).

Motivated by the above-mentioned issues, we provide an overview of existing predictability measures developed for various types of time series (univariate, multivariate, and categorical) as well as for network links. Moreover, we conduct an analysis of the correlation rate between intrinsic and realized predictability measures, utilizing both artificial and real-world time series. Our findings reveal statistically significant correlations between specific pairs of these measures. In short, the impact of this study is as follows:We formalize the predictability measure concept, specifying measures of realized and intrinsic predictability;We provide a comprehensive survey of predictability measures for time series and network links, encompassing both realized and intrinsic predictability measures;We provide the correlation analysis results between intrinsic and realized time series predictability measures, offering valuable insights for assessing achievable forecast accuracy levels prior to the forecasting moment.

## 2. Overview of Predictability Measures

The task of predicting the future has become a fundamental practice for many scientists across various disciplines. As a result, a multitude of forecasting methods have been devised, catering to a diverse array of objects and phenomena. This extensive range of objects has led researchers to explore predictive techniques in fields as varied as economics, climate science, medicine, social sciences, and more.

Among the various entities whose behaviors have been subject to prediction, time series [1,2] and network links [3,4] stand out as some of the most prominent and extensively studied. Time series, which represent a sequence of data points indexed by time intervals, find applications in fields such as finance [13], meteorology [14], and stock market analysis [15], where forecasting future values based on past observations is of paramount importance. Network links, on the other hand, are crucial in understanding and modeling the relationships and interactions among entities in complex systems, such as social networks [16], transportation networks [17], and biological systems [18].

We can highlight certain methods that are widely utilized across diverse fields. Among these, linear and nonlinear regression models stand out as some of the most straightforward techniques. What makes them particularly appealing is their simplicity, as they do not demand significant computational resources and can be constructed without reliance on specialized tools or software. Regression models continue to find applications in research and practice, as evidenced by studies such as [5,19,20].

The autoregressive model ranks among the most popular forecasting methodologies, largely due to its well-defined algorithmic framework for model construction and parameter selection [21]. The research has seen extensive adoption of various autoregressive models, such as ARMA, ARIMA, ARCH, GARCH, and others, to predict a wide range of phenomena, such as market prices [15,22,23,24], network traffic [25,26,27], and social processes [28,29,30]. Oftentimes, hybrid autoregressive models that combine multiple forecasting techniques are used [31,32,33,34]. By combining the strengths of different methodologies, these hybrid approaches seek to harness the complementary predictive capabilities of various models, yielding more accurate and robust forecasts.

Neural networks represent a specific type of nonlinear functional architecture that involves iteratively processing linear combinations of inputs through nonlinear functions [35]. Artificial neural networks have found extensive application in various domains, demonstrating their efficacy in predicting stock prices and indices [6,36,37], addressing industrial challenges [38,39,40], facilitating medical forecasts [41,42,43,44,45], enhancing weather forecasts [46,47,48,49], and tackling a myriad of other challenges [50,51,52].

Well-established classification methods, including decision trees, k-Nearest Neighbors (k-NN), and Support Vector Machines (SVM), among others, have demonstrated their applicability in predicting links within a network [7], achieving competitive levels of accuracy. The utilization of these methods in link prediction tasks allows researchers to harness their robustness and adaptability to different datasets and graph structures. In the work of [53], a comparative analysis is conducted on various node similarity measures, encompassing both node-dependent indices and path-dependent indices. These measures serve as essential components in link prediction, aiding in quantifying the potential connections between nodes and guiding the predictive modeling process. Beyond these widespread techniques, there exist link prediction methods that leverage concepts such as random walks [8,54], matrix factorization [55,56], and others [57,58,59]. These methods are based on diverse mathematical frameworks to capture complex patterns within the network’s topology and provide valuable insights into the potential links between nodes.

The approaches discussed in the referenced papers are oriented towards the development of methods capable of making predictions with the desired level of quality on test datasets. However, it has been observed that authors often overlook certain questions: What is the overall predictability of this object? Can a model be devised that exhibits superior performance on this dataset?

To validate this concept, we introduce a citation network comprised of the regarded papers focused on forecasting and predictability assessment methods for various objects, such as time series and network links. The network and its connections are depicted in Figure 1. Notably, the figure illustrates that research studies aimed at enhancing forecast accuracy are rarely linked to predictability studies.

In this section, we aim to clarify the distinction between prediction accuracy and predictability. We provide a comprehensive survey within the field of predictability and the measures employed for its assessment.

### 2.1. General Predictability Concepts

To evaluate the quality of predictive models on data, methods of predictability analysis for the modeling object can be employed. According to [11], the concept of object predictability should be divided into two components: realized predictability (RPr) and intrinsic predictability (IPr). The first type of predictability, realized predictability, is a function that depends on the forecasting model employed:(1)$$\rho_{R}=\rho \left(m,S\right),$$
where *m* is the forecasting model and *S* is the object or class of objects. For example, different forecast quality metrics, such as MSE, MASE, RMSE, etc., are nothing but realized predictability measures. The second type, intrinsic predictability, is independent of the model used:(2)$$\rho_{I}=\rho \left(S\right),$$
where *S* is the object or class of objects. Calculating theoretical estimates of intrinsic predictability for objects can be challenging, even in the case of simple data classes [60]. Therefore, the upper bound of realized predictability, which represents the predictability achieved by an optimal model, is used to estimate intrinsic predictability:(3)$${\hat{\rho }}_{I}=\underset{m}{\sup}\rho \left(m,S\right).$$

Low values of realized predictability could indicate not only the inherent complexity of the modeled object but also potentially inadequate quality of the selected model. Conversely, intrinsic predictability is independent of the forecasting model and can, therefore, be utilized to assess the data quality.

One of the central questions that motivated the experimental aspect of this study is how to establish a connection between quality measures (RPr), $\rho_{R}$, and a measure of intrinsic predictability, $\rho_{I}$. While we possess distinct tools for predictability estimation, the fundamental question is: do they truly have a meaningful relationship with each other? We will discuss this later in the paper.

Researchers have developed various approaches to measure the predictability of diverse objects, including univariate and multivariate time series, categorical time series (event sequences), network links, and more. It is important to note that there is not a singular methodological approach in this field, as different researchers approach this issue from the perspectives of various scientific domains, such as information theory, dynamical system theory, approximation theory, and others. In the following sections, we provide an overview of the existing approaches used to assess the predictability of different types of objects.

### 2.2. Time Series Predictability

Prior to discussing time series predictability measures, we focus on the notion of intrinsic unpredictability (IPr) concerning a random variable. The definition of an unpredictable random variable was introduced in reference [61]. A random variable $\xi_{t}$ is deemed unpredictable with respect to an information set $\Omega_{t-1}$ if the conditional distribution $F_{\xi_{t}}\left(\xi_{t}|\Omega_{t-1}\right)$ aligns with the unconditional distribution $F_{\xi_{t}}\left(\xi_{t}\right)$ of $\xi_{t}$, that is:(4)$$F_{\xi_{t}}\left(\xi_{t}|\Omega_{t-1}\right)=F_{\xi_{t}}\left(\xi_{t}\right).$$

Specifically, when $\Omega_{t-1}$ comprises past realizations of $\xi_{t}$, Equation (Equation 4) suggests that having knowledge about these past realizations does not enhance the predictive accuracy of $\xi_{t}$. It is important to note that this form of unpredictability in $\xi_{t}$ is an inherent attribute, unrelated to any prediction algorithm.

#### 2.2.1. Univariate Time Series

Methods for estimating univariate time series predictability can be categorized into two groups: methods for estimating sample predictability and methods for estimating intrinsic predictability. The first group of methods aims to evaluate realized predictability (RPr) and encompasses measures that analyze forecast quality. The second group comprises intrinsic predictability (IPr) estimation methods, which are often rooted in information theory approaches, particularly those involving Shannon entropy. We will now delve deeper into each of these groups.

The starting point for developing sample predictability measures is the work of [62], in which the predictability of a time series generated by a wide-sense stationary process is proposed to be evaluated as the ratio of the theoretical optimal forecast and the original time series variances. Consequently, computing predictability values requires knowledge of the optimal forecast, which is challenging to obtain for real-world time series. Therefore, coefficients for the sample predictability estimation (RPr), based on the approach from [62], are developed [63]. The approach from [62] assesses predictability as the ratio of the optimal forecast and the original series variances, but instead of using the optimal forecast error, while the proposed approach from [63] employs the forecast error shown by a specific model. Thus, the coefficient of efficiency (CE) indicates the ratio of the sample variance of the model’s forecast error to the sample variance of the time series, measured over the entire observation period. The seasonally adjusted coefficient of efficiency (SACE) differs from CE in that the series variance is considered within a specific season. The SACE coefficient is useful for time series where the mean value changes based on the season.

Furthermore, this group includes the work [64], which introduces a metric of realized predictability based on the ratio of two values: the sum of squared errors in the case of forecasting the original time series and the case of forecasting the same series after random shuffling. It is evident that measuring the predictability of a series using the three mentioned measures is entirely dependent on the model’s performance, and therefore, does not provide insight into the intrinsic predictability of the data.

The second group of methods for analyzing the predictability of univariate time series is based on (but not limited to) estimating various forms of entropy for the series, thereby allowing the assessment of data properties rather than models. This group of methods, in turn, is further divided into two subgroups: methods for assessing the predictability of the entire series and methods for assessing predictability on specific scales. Different scales of the series refer to series obtained from the original by retaining only those values that occur at specific intervals of time. Investigating the predictability of the series at different scales can provide valuable information about long-term correlations in the data, which can subsequently be utilized in training predictive models.

In [10], the Shannon entropy formula is employed to assess the intrinsic predictability of clients’ trajectories as time series composed of their coordinates. Additionally, utilizing Fano’s inequality, which relates the average information loss in a noisy communication channel to the probability of errors during signal reception, the authors established an upper bound for the trajectory’s predictability. However, computing the trajectory entropy value using the Shannon formula necessitates the calculation of the probability of finding a subsequence within the original trajectory for all potential subsequences. This requires substantial time and computational resources. Moreover, for time series containing observations spanning several decades, such calculations would be practically infeasible. For this reason, the Lempel–Ziv–Welch data compression algorithm is applied to estimate the real entropy value. The core concept of this algorithm involves dynamically building a phrase dictionary by sequentially reading the text character by character and comparing the resulting character sequences with the dictionary entries. Consequently, an approximate entropy value for a time series $\left(y_{1},y_{2},\ldots y_{N}\right)$ can be computed by iterating through all possible subsequences that start with an element at a specific index and are not continuous subsequences of trajectory elements whose indices precede the given index:(5)$$\mathrm{Ent}={\left(\frac{1}{N}\sum_{i}\Lambda_{i}\right)}^{-1}\ln\left(N\right),$$
where $\Lambda_{i}$ is minimum length *k*, such that the subsequence starting from *i* with length *k* does not appear as a continuous subsequence of $\left(y_{1},y_{2},\ldots y_{i-1}\right)$.

The authors of [65] made slight modifications to the formula for approximating entropy computation and also improved the algorithm for iterating through trajectory subsequences. A series of experiments described in [65] demonstrates that the entropy estimation proposed by the authors yields results that are closer to the theoretical predictability bound compared to the estimation presented in [10].

In Reference [66], permutation entropy (PE) is proposed for assessing the intrinsic predictability of a series $\left(y_{1},y_{2},\ldots y_{N}\right)$. Permutation entropy can be viewed as a method for analyzing the complexity of a time series. It is also rooted in the Shannon entropy formula, which is based on the probabilities of encountering permutations in certain subsequences of the series. The mapping, operating from a space composed of subvectors (embeddings) of the original series, into the space of permutations, is established through the relationship of order among adjacent elements of the subvectors. More formally, to calculate the PE value, one should consider the original series embedding of the dimension *m* with a time delay τ, which consists of the following vectors: $Y_{j}^{m,\tau }=\left(y_{j},y_{j+\tau },\ldots y_{j+(m-1)\tau }\right)$, where j∈{1,2,…,N−(m−1)τ}. Next, each of the M=N−(m−1) vectors is mapped to one of the m! possible permutations. Finally, the equation for calculation the permutation entropy is as follows:(6)$$\begin{array}{c} \mathrm{PE}\left(m,\tau \right)=-\sum_{i:\pi_{i}^{m,\tau }\in \Pi }P\left(\pi_{i}^{m,\tau }\right)\cdot \ln\left[P\left(\pi_{i}^{m,\tau }\right)\right],\\ P\left(\pi_{i}^{m,\tau }\right)=\frac{\sum_{j\leq M}I_{u:type\left(u\right)=\pi_{i}}\left(Y_{j}^{m,\tau }\right)}{\sum_{j\leq M}I_{u:type(u)=\Pi }\left(Y_{j}^{m,\tau }\right)}, \end{array}$$
where $\Pi ={\left\{\pi_{i}^{m,\tau }\right\}}_{i=1}^{m!}$ is the set of all possible permutations of *m* elements, type(·) is the relation that maps the vector $Y_{j}^{m,\tau }$ to the permutation $\pi_{j}^{m,\tau }$, and $I_{A}$ is the characteristic function of the set *A*.

There are also methods for measuring the complexity of a time series that take into account not only the order in which its elements are arranged, but also the amplitude of the series values. To incorporate the amplitude of the series values in assessing its complexity, weighted permutation entropy (WPE) was developed [67]. The computation logic of WPE is similar to that of PE, with the distinction that each permutation is associated with a weight equal to the sample variance of the corresponding subvectors of the original series.

In Reference [68], the relationship between the predictability of a time series, expressed through weighted permutation entropy, and the Mean Absolute Scaled Error (MASE) is investigated. Experiments are conducted on artificially generated time series with explicitly defined structures: short-term correlation (AR(1) model), long-term correlation (ARFIMA(0, d, 0) model), multifractal time series (binomial multifractal model), and chaotic time series (Lorenz attractor). As a result, predictability bounds are derived for time series with different structures. By using these bounds to determine the structures of real-world time series, the authors achieve improved forecasting quality by employing models that are most suitable for time series with specific structures.

Another measure of intrinsic predictability for a time series, based on entropy, is the Wavelet Energy Entropy Measure (WEEM) [69]. Wavelet transformation converts a signal (time series) from its time-domain representation into a frequency-time representation, making certain aspects of the original signal (time series) more amenable to study. The authors note that the novelty of their proposed measure lies in its ability to account for the dynamics of the process generating the time series across different scales during its computation. To calculate the WEEM measure, it is necessary to generate a time series consisting of white noise realizations, with a length matching that of the original series. Then, the energy of the wavelet transformation of the original series should be computed, and this value will be used to calculate the continuous entropy of the wavelet transformation. This entropy should also be computed for a time series composed of white noise values, and then both entropy values will be combined in the formula for calculating WEEM. The comparison with the entropy of white noise is explained by the fact that the entropy of white noise is maximal, as its energy distribution is uniform across any interval.

In Reference [70], a method is introduced for quantifying the irregularity of the time series by using the spectrum entropy (SE). It serves as a mean of quantifying the irregularity present in time series due to the fact that spectral entropy reflects the relative sharpness or uniformity of the spectral distribution. SE mainly depends on spectral variables, such as the degree of dominance of a few peaks, the number of peaks, and their peakedness. As the degree of time series irregularity increases, the SE value also increases.

The obtained power spectrum S(f) of the time series $\left(y_{1},y_{2},\ldots y_{N}\right)$ is normalized using the following equation: $\sum_{f=0}^{f_{n}}S(f)=1$, where $f_{n}$ is the sampling frequency. The entropy of the relative power (normalized power) across the entire frequency range is calculated as follows:(7)$$\mathrm{SE}=\sum_{f=0}^{f_{n}}S\left(f\right)\cdot \log_{2}S(f).$$

In [71], the authors define a measure of dynamic complexity known as the entropy of the singular value decomposition (SVD entropy). SVD entropy serves as an indicator of the number of eigenvectors required for a sufficient explanation of the data. The authors note that the method does not require a significant computational overhead and can be employed in real-time systems.

To calculate the SVD entropy value for a time series $\left(y_{1},y_{2},\ldots y_{N}\right)$, we consider the embedding matrix *X* for this series, which can be written as follows:(8)$$X={\left[x\left(1\right),x\left(2\right),\ldots x\left(N-\left(order-1\right)delay\right)\right]}^{T},$$
where $x\left(i\right)=\left[y_{i},y_{i+delay},\ldots ,y_{i+(order-1)delay}\right].$ The entropy of the singular value decomposition is then defined as:(9)$$\mathrm{SVD}\_\mathrm{ent}=-\sum_{i=1}^{M_{s}}\bar{\sigma_{i}}\log\bar{\sigma_{i}},$$
where $M_{s}$ is the number of singular values of the embedding matrix *X* and $\bar{\sigma_{1}},\bar{\sigma_{2}},\cdots ,\bar{\sigma_{M_{s}}}$ are the normalized singular values of *X*, calculated as $\bar{\sigma_{i}}=\sigma_{i}/\sum_{i^{\prime }}\sigma_{i^{\prime }}$.

Furthermore, in [72], the authors point out that entropy, as a measure of uncertainty, lacks sensitivity to nonstationarity in the signal. To address this limitation, they utilized time-dependent entropy (TDE) based on a sliding temporal window technique.

All the mentioned approaches to measuring the predictability of a series are aimed at analyzing the entire series. However, it can be useful not only to answer the question of how predictable a particular series is, but also to analyze in what scale this series exhibits high or low predictability (IPr).

In [73], one of the first predictability measures, which analyzes not the entire time series but rather its subvectors of varying lengths, is introduced. This measure is known as Approximate entropy, which serves as a method to quantify the degree of regularity and unpredictability of fluctuations within a time series. The algorithm for computing the Approximate entropy value for a time series $\left(y_{1},y_{2},\ldots y_{N}\right)$ with fixed parameters *m* (chosen length of subvectors) and *r* (chosen filtering level or neighbourhood radius) consists of the following steps:Form a sequence of vectors x(1),x(2),…,x(N−m+1) in $R^{m}$, a real *m*-dimensional space, defined by $x\left(i\right)=\left[y_{i},\ldots ,y_{i+m-1}\right]$;Use the sequence x(1),x(2),…,x(N−m+1) to construct, for each *i*, 1≤i≤N−m+1, $C_{i}^{m}\left(r\right)=\left\{\mathrm{number}\;\mathrm{of}\;x\left(j\right)\;\mathrm{such}\;\mathrm{that}\;d\left[x\left(i\right),x\left(j\right)\right]\leq r\right\}/\left(N-m+1\right)$;Define $\Phi^{m}\left(r\right)={\left(N-m+1\right)}^{-1}\sum_{i=1}^{N-m+1}\ln C_{i}^{m}\left(r\right)$;Calculate the Approximate entropy value:
(10)$$\mathrm{App}\_\mathrm{ent}\left(r,m\right)=\Phi^{m}\left(r\right)-\Phi^{m+1}\left(r\right).$$

Sample entropy, proposed in [74], expresses the conditional probability that two samples (sequences) will remain identical after adding a new point to each sample. Essentially, Sample entropy is a modification of Approximate entropy. Higher values of Sample entropy suggest greater complexity, while smaller values indicate more self-similarity and regularity in time series. For a time series $\left(y_{1},y_{2},\ldots y_{N}\right)$, the Sample entropy value with fixed parameters *m* (chosen length of subvectors) and *r* (chosen filtering level or neighbourhood radius) can be found by the following expression:(11)$$\mathrm{Samp}\_\mathrm{ent}\left(r,m\right)=-\ln\left(\frac{\Phi^{m+1}\left(r\right)}{\Phi^{m}\left(r\right)}\right).$$

The authors of [74] conducted a series of experiments in which they compared Approximate entropy with the proposed measure, Sample entropy. By presenting several arguments that highlight the advantages of the proposed measure, the authors concluded that Sample entropy offers an enhanced assessment of time series regularity.

Furthermore, for the analysis of predictability in different scales, approaches based on multiscale entropy are employed. Multiscale Entropy (MSE) [75] is widely used for assessing the predictability of a time series within different scales. The algorithm for calculating MSE involves two main steps: (a) the procedure of constructing a set of time series at different scales (the coarse-graining procedure) and (b) the computation of Sample entropy [74] for the time series at various scales. Sample entropy can only take two values, namely 0 and 1. As a result, even slight changes in the behavior of the series can lead to sharp changes in the values of multiscale entropy. To address this issue, Flexible Multiscale Entropy (FMSE) [76] is proposed, with values ranging from 0 to 1, enhancing the reliability and stability of time series predictability measurement. Another modification of MSE, Composite Multiscale Entropy (CMSE) [77], is previously suggested to enhance the stability of the MSE method. The results of experiments from [76], conducted on both synthetic and real time series, demonstrate that FMSE exhibits more consistent behavior across various scales than MSE and CMSE; FMSE’s greatest superiority is observed in short series. Moreover, in [78], multiscale entropy differences (MED) are proposed to evaluate the predictability of financial time series.

As mentioned earlier, different researchers approach the question of predictability assessment from various scientific perspectives. In [9], five measures (IPr) are identified that characterize the dynamic system generating a specific time series: correlation dimension, correlation entropy, Kolmogorov–Sinai entropy, Hurst exponent, and noise measure. The authors computed the values of all these measures for one-dimensional time series from a dataset and subsequently conducted series clustering based on the obtained measure values. After computing the prediction errors for series in different clusters, the authors concluded that the resulting clusters exhibit varying levels of realized predictability.

#### 2.2.2. Multivariate Time Series

The goal of assessing the predictability of components within a multivariate time series (also referred to as features) is to select a set of features that best describe the behavior of the object and enable the model to make forecasts of the desired quality. In fact, there may be situations where using a particular feature as a predictor does not improve the forecast quality, complicating the search for patterns in the data. When dealing with large volumes of input data, there arises a need to match the original features with a specific subset of smaller-sized features that can make the model’s forecasts more stable and effective. However, standard dimensionality reduction methods are focused on preserving data properties unrelated to predictability, which introduces the risk of losing important information contained within the data [79].

There is an entire group of methods [79,80,81,82] in the field of extracting predictable features from multivariate time series. All works within this group share a similar algorithm. The input consists of multivariate time series, and the objective is to find a mapping from a space whose dimension equals the number of series components to a lower-dimensional space. The principle of dimensionality reduction may vary depending on the algorithm, yet all methods boil down to an optimization problem. The approach [82] is aimed at theoretically assessing realized predictability, while the other works within this group deal with the intrinsic predictability of features.

Indeed, in [80], a method for extracting slowly changing (or invariant) features, known as Slow Feature Analysis (SFA), is proposed. This method is utilized to analyze multivariate time series containing sensor data. The principle of feature extraction for creating a lower-dimensional space involves retaining those features that change over time as slowly as possible. Ultimately, the dimensionality reduction task is reduced to a variational calculus optimization problem.

The Forecastable Component Analysis (ForeCA) method [81] also addresses the task of feature selection, in this case, forecastable features. The work introduces a measure of predictability for time series generated by stationary random processes. The calculation of this measure employs the entropy of the process, which, in turn, is determined using spectral density. The ultimate task of finding features is reduced to maximizing this predictability measure.

A similar task of finding a set of features is considered in [82] (Predictable Feature Analysis, PFA), with the distinction that feature selection is carried out considering a specific model (i.e., the features that are well predicted by it). A criterion is derived that the model must adhere to in order to be used with PFA. It is worth noting that despite analyzing a specific model, this method examines theoretical predictability without utilizing forecast results. The advantage of this approach is the knowledge of a certain model that is able to make the predictions of the desired quality. However, the optimization problem is more challenging than in SFA: the forecasting optimization problem is embedded within the optimization problem of searching for predictable features.

The method from [79] (Graph-based Predictable Feature Analysis, GPFA) is based on interpreting predictability as a situation where the variance of a time series in the next time step is small given that the current value of the series is known. The dimensionality reduction task is formulated as the search for an orthogonal transformation of the original series. The term graph in the method’s name is mentioned simply because it is used as an auxiliary tool to search for the columns of the orthogonal transformation matrix. Additionally, the predictable components of a multivariate series can be seen as neighboring nodes on a specific graph, connected by an edge.

All the methods discussed in this section (apart from [82] that theoretically assesses realized predictability) are aimed at estimating intrinsic feature predictability. Additionally, there exists a straightforward approach to estimate realized feature predictability [83,84]. This approach identifies the most useful features for predicting the remaining time of system performance. The predictability measure is defined as a function dependent on the prediction horizon, model class, model parameters, and the required accuracy threshold. The proposed predictability measure combines the threshold and the accuracy achieved by the model into a single value ranging from 0 to 1. Subsequently, pairs (a set of features and a model) with a favorable predictability value are selected through brute force.

#### 2.2.3. Categorical Time Series

This section is dedicated to the predictability of (often short) univariate time series consisting of categorical values. In practice, these time series can be referred to as events or event sequences. For instance, in [85], sequences composed of items viewed or purchased by users during a single session on a retailer’s website are considered. Predictability in this context refers to the probability of correctly determining the next element in the sequence (i.e., the purchase of a specific item), given the session’s start and the user’s session history. The authors provide an estimation of the maximum theoretical predictability of the sequence, expressed using entropy as formulated in [10]. Furthermore, the theoretical predictability realized by specific algorithms (RPr) is assessed by analyzing potential algorithm outcomes and selecting the result with the best forecasting quality. For instance, in the case of a Markov chain model, predictability is evaluated as the proportion of observations where the most likely transition from one state to another occurred. Thus, theoretical predictability can only be assessed for explicit models. Estimating the theoretical predictability for black-box models using this approach is not feasible.

In Reference [86], the authors consider the problem of forecasting the next point in a trajectory (the category of the point, not its coordinates). Similar to [86], the maximum theoretical predictability of the sequence is assessed using the entropy formula from [10]. Additionally, two statistics are introduced to measure the gap between theoretical predictability (IPr) and the maximum prediction accuracy achieved using a set of models (Equation 3).

In [87,88], the authors assess the realized predictability (RPr) of client’s transactional sequences by employing a coefficient based on the mean absolute error of the selected predictive model for each sequence. Subsequently, they categorize all sequences into predictability classes based on the values of the predictability measure. This approach can be utilized to gauge the predictability level of a sequence prior to forecasting, by utilizing a form of meta-classifier that assigns categorical time series to their corresponding predictability classes. Experiments demonstrated the efficiency of this approach, as the estimated predictability classes consistently align with those obtained through the application of a prediction model.

### 2.3. Network Link Predictability

Most of the studies on network link predictability discussed in this section are focused on assessing intrinsic predictability. In Reference [12], a network is considered predictable if the removal or addition of a small number of randomly chosen nodes preserves its fundamental structural characteristics. Such networks are referred to as structurally consistent. The proposed measure is the Universal Structural Consistency Index, which is based on perturbing the adjacency matrix and evaluates the corresponding changes in network structural features. Through conducted experiments, a strong correlation is revealed between link prediction accuracy and the structural consistency index in various real-world networks, demonstrating the applicability of network structural consistency as a link predictability assessment. Moreover, this index can be used in tasks of missing links prediction. Such experiments with networks constructed using the Erdos–Renyi model indicate, as expected from the networks’ construction, that this type of networks is poorly predictable.

Furthermore, to assess the predictability of network structure, the normalized shortest compression length of the network structure can be employed [89]. Any network can be transformed into a binary string through compression. The length of the string increases as the randomness in the network structure grows. The authors compared their proposed predictability measure with the accuracy of the best available link prediction algorithm (as an approximation of the optimal algorithm) estimated via performance entropy and find a strong correlation.

Another work focusing on the assessment of the intrinsic predictability of network links is [60]. By considering ensembles of well-known network models, the authors analytically demonstrated that even the best possible link prediction methods provide limited accuracy, quantitatively dependent on the ensemble’s topological properties such as degree heterogeneity, clustering, and community structure. This fact implies an inherent limitation on predicting missing links in real-world networks due to uncertainty arising from the random nature of link formation processes. The authors show that the predictability limit can be estimated in real-world networks and propose a method to approximate this limit for real-world networks with missing links. The predictability limit serves as a benchmark for evaluating the quality of link prediction methods in real-world networks. Additionally, the authors conducted experiments comparing their proposed predictability measure with the structural consistency index from [12].

The authors of [90] assessed the predictability of links in temporal networks. The temporal nature of links in many real-world networks is not random, but predicting them is complicated due to the intertwining of topological and temporal link patterns. The paper introduces an approach based on Entropy rate, which combines both topological and temporal patterns to quantitatively assess the predictability of any temporal network (in previous works, only temporal aspects were considered). To examine both topological and temporal properties of the network, the sequence of adjacency matrices is treated as realizations of a random process. The subsequent procedure is similar to the one in [10] for trajectory predictability estimation: the entropy rate and theoretical upper bound of intrinsic predictability are derived, and then applying the Lempel–Ziv–Welch data compression algorithm yields an expression for the approximate entropy value. It is noted that for most real-world temporal networks, despite the increased complexity of predictability estimation, the upper bound of combined topological-temporal predictability is higher than that of temporal predictability.

Furthermore, there are two studies [91,92] focusing on the realized predictability of network links; specifically, the predictability observed through a selected feature-based link prediction model. The authors evaluate link predictability by assessing the error of the chosen model and divide the links within a small portion of the network into high and low predictability classes based on the error value. Subsequently, they train a meta-classifier on this subset of the network to estimate the predictability class using certain link features. This meta-classifier can then be applied to the entire network to estimate link predictability without the time-consuming process of training a link prediction model.

Moreover, there are methods that involve converting time series into networks. These methods can be categorized into three classes based on the type of resulting network [93]: (a) proximity networks; (b) visibility networks; (c) transition networks. The first category of methods constructs networks by utilizing information about the mutual proximity of various segments within time series. Mutual proximity can be measured in various ways, such as through the correlation between time series cycles (resulting in a cycle network [94]), the correlation between time series segments (resulting in a correlation network [95]), or the closeness of time series segments in phase space (resulting in a recurrence network [96]). The second category, known as visibility network methods, generates networks based on the convexity of consecutive observations in series [97]. Lastly, the class of methods producing transition networks constructs networks by considering the transition probabilities between groups of aggregated values from series [98].

However, to the best of our knowledge, there is only one study [99] that analyzed predictability under such transformations. The authors utilized the transition network approach to convert time series into networks. They then calculated network characteristics and employ them for clustering time series into two groups. By measuring the forecasting errors obtained by various models on time series from these clusters, they concluded that these clusters can be considered as classes of high and low predictability, as the mean forecasting error in one class is significantly lower than in the other.

### 2.4. Overview Summary

To assess the predictability of data, there is a multitude of measures that allow working with various objects: univariate and multivariate time series, categorical time series (sequences of events), and network links. In the case of time series, measures are developed to evaluate the predictability of the series as a whole, as well as at specific scales. The diversity of measures presented in the studies is due to researchers expressing their perspectives on predictability assessment from various scientific domains, such as information theory, dynamical systems, approximation theory, and so on. However, despite the variety of existing predictability measures, the challenge of proper analysis of connection between intrinsic and realized predictability remains unresolved. In the next section, we will try to shed a light on this problem.

## 3. Correlation Rate between Intrinsic and Realized Predictability

The objective of the second part of the paper is to investigate the correlation between intrinsic and realized predictability. Exploring the presence of a connection between the measure of forecast accuracy and the measure of intrinsic predictability proves valuable in addressing the challenge of evaluating forecast quality prior to the forecasting moment. It is worth noting that similar work was conducted in [11], focusing on a single intrinsic predictability measure (permutation entropy). In contrast, our study offers a more comprehensive approach by encompassing six distinct intrinsic predictability measures through an extensive experimental investigation.

### 3.1. Method Description

#### 3.1.1. Problem Statement

The problem can be formulated as follows: consider a set of *N* time series of a certain length, denoted as ${\left\{y_{i}\right\}}_{i=1}^{n}$, along with *m* forecasting models, represented as ${\left\{m_{j}\right\}}_{j=1}^{M}$. For each series $y_{i}$, it is possible to compute the measure of intrinsic predictability denoted as $\rho_{I_{i}}=\rho \left(y_{i}\right)$. Utilizing the model $m_{j}$ to forecast the time series $y_{i}$, the measure of realized predictability, labeled as $\rho_{R_{j,i}}=\rho \left(m_{j},y_{i}\right)$, can be computed. This measure will not solely rely on the series $y_{i}$, but also on the specific forecasting model $m_{j}$ chosen. By setting, for instance, j=1 (using a specific forecasting model $m_{1}$), what will be the correlation $corr(\rho_{I_{i}},\rho_{R_{1,i}})$? Moreover, when analyzing the entire array of forecasting models, what will be the predictability measures’ correlation $corr(\rho_{I_{i}},\rho_{R_{j,i}})$? The insights derived from these inquiries will offer valuable insights for evaluating the internal complexity of the series.

#### 3.1.2. Pipeline

The task of examining the correlation between intrinsic and realized predictability is accomplished through four key steps (as illustrated in Figure 2). In the initial step, known as data processing, alongside common transformations such as addressing missing values, scaling, and partitioning into training and test sets, the time series are adapted to enable the application of various forecasting models. It is important to mention that the experiments are conducted using both real-world time series data and synthetic ones, necessitating the generation of time series at this stage as well.

Utilizing the processed data, measures of intrinsic predictability are computed (six measures are highlighted in Figure 2). Concurrently, the time series are fed into the forecasting models (five models are employed, also depicted in Figure 2). Subsequent to applying the forecasting models to the time series, realized predictability measures (five forecast error metrics) are computed based on the generated forecasts. In the concluding step, all the computed datasets are gathered, and pairwise correlation coefficients are computed, forming correlation matrices.

#### 3.1.3. Forecasting Models and Predictability Measures

As mentioned in Section 3.1.2, various forecasting models are employed to investigate the correlation between measures of intrinsic and realized predictability.

ARIMA, or Autoregressive Integrated Moving Average, is a widely used forecasting model for predicting time series across a range of fields. One of the strengths of this model is its clear mathematical foundation, accompanied by a step-by-step algorithm for constructing the model and selecting its parameters [21]. ARIMA comprises three parameters (p,d,q), where *p* is the autoregressive order, *q* is the moving average order and *d* is the difference order. ARIMA(p,d,q) serves as an extension of ARMA(p,q) models to address nonstationary time series. These nonstationary series are transformed into stationary ones by taking differences of order *d* from the original time series—a process known as integration.

The ARIMA methodology for time series involves initially assessing the stationarity of the series. The degree of integration of the time series is determined, typically restricted to either first or second order. If the degree of integration surpasses zero, the series is transformed through differencing by the corresponding order, and subsequently, an ARMA(p,q) model is built.

In the following experiments, the model parameters are determined using the grid enumeration method. This involves considering all combinations of p∈{1,5}, d∈{0,2}, q∈{1,5}, and selecting the parameter sequence that minimizes the Akaike criterion.

LSTM (Long Short-Term Memory) is a type of recurrent neural network designed to capture long-term dependencies within data sequences. The LSTM model was introduced in [100] and continues to be extensively utilized, particularly in tasks that involve retaining information over extended time intervals.

The LSTM architecture employed for predicting the next observation in the time series comprises a single hidden layer of 50 LSTM units and an output layer tasked with predicting a single numerical value. The input shape is determined by the number of time steps and the number of features. In this context, the number of time steps is set to 10 (further elaboration on the selection of time steps is provided). Given that the experiments utilize univariate series, the number of features is one. To train the model, the Adam optimization algorithm and the Mean Squared Error (MSE) loss function are employed.

Two tree-based machine learning algorithms are applied: Random Forest and XGBoost (eXtreme Gradient Boosting). Decision trees are a supervised learning method that works by recursively splitting the input data into subsets based on the values of different features. Each split is determined by a decision rule, which is chosen to maximize the separation of the data into different classes or to minimize the prediction error.

Random Forest is an algorithm for solving classification and regression problems based on decision trees and ensemble learning. In fact, the algorithm is based on bagging, so all trees of the ensemble are constructed in parallel on independent samples.

Similar to Random Forest, Gradient Boosting is based on ensemble learning, but in this case, the type of ensemble method is boosting. This means that each tree in the ensemble is constructed in such way as to focus on instances where the previous tree failed. Thus, one tree learns from the mistakes of another tree.

In this study, the Random Forest algorithm is employed with the following parameters: an ensemble consisting of 1000 trees, and the squared error function is utilized to evaluate the quality of splits. Similar to the LSTM model, the input shape consists of the number of time steps and the number of features, following similar data processing steps. XGBoost is also employed, utilizing 1000 gradient boosted trees, with the squared error serving as the corresponding metric for performance evaluation.

Convolutional Neural Networks (CNNs), originally developed for image data, are increasingly being applied to address time series forecasting problems. As a result, we also incorporate this model into our study. A typical CNN consists of three types of layers: a convolutional layer, a pooling layer, and a fully connected layer. However, the architecture can vary depending on the specific application and design choices.

The CNN model used in this study follows the architecture outlined below. The initial layer is a convolutional hidden layer that operates over sequences. This layer consists of 64 filter maps, and the kernel size is set to two. The ReLU activation function is applied. Following the convolutional layer, a pooling layer is employed to filter the output of the previous layer, capturing the most important features. Subsequently, a flatten layer is used to transform the feature maps into a single one-dimensional vector. A fully connected dense layer interprets the features extracted by the convolutional segment of the model. The final dense layer is designed to predict a single value, representing an observation from the time series. Similar to the LSTM model, the Adam optimization algorithm and MSE-loss function are utilized for model fitting.

Entropy and its various modifications are employed to assess the intrinsic predictability of the series, namely, Entropy estimated via Lempel–Ziv data compression (Equation 5), Permutation entropy (Equation 6), Spectral entropy (Equation 7), Singular value decomposition entropy (Equation 9), Approximate entropy (Equation 10), and Sample entropy (Equation 11). Detailed descriptions and formulas can be found in Section 2.2.1. Forecast errors serve as measures of realized predictability, utilizing five standard metrics: MAE, MSE, RMSE, MAPE, and $R^{2}$. While these measures are standard, their descriptions and calculation formulas are provided in Table 1.

It is worth mentioning that measures of intrinsic predictability are computed on the training set, whereas forecast errors are computed on the test set after the model’s predictions are obtained. The assumption is that by calculating measures such as entropy for the series, we can gain additional insights into the intrinsic complexity before applying the forecasting model.

### 3.2. Experiments and Results

#### 3.2.1. Data Description, Generation, and Processing

To ensure the reliability of the results, both real-world and artificial series are utilized in the experiments. Artificial series are created by combining various components, such as periodic components, white noise, and random walks, in different proportions. The general equation for the generated series is as follows:(12)$$y_{i}=k_{t}Y_{t}+k_{p}Y_{p}+k_{n}Y_{n}+k_{r}Y_{r},$$
where $k_{t}$, $k_{p}$, $k_{n}$, and $k_{r}$ are coefficients, $Y_{t}$ corresponds for values from uniform distribution, $Y_{p}$ is the periodic component of the time series, $Y_{n}$ is the random component of the time series, and $Y_{r}$ is the random walk component. Examples of the generated series are shown in Figure 3. A total of 250 generated series are produced for the experiments. The code for the artificial time series generation (the length of time series as well as their entity can be customized) is available on GitHub (https://github.com/Anthony-Cov/Timesergraph/blob/main/ArtSerGenerator.py, accessed on 7 September 2023).

To assess the correlation on real-world time series, another set of 250 series is collected, reflecting the dynamics of various socio-economic indicators. This dataset primarily includes stock prices of various companies, along with dynamics of consumer loans, crude oil prices, deposits, unemployment rates, capital expenditures, inflation rates, and many others. Part of the series is available on our GitHub (https://github.com/Anthony-Cov/Timesergraph/tree/main/RealWeekly, accessed on 7 September 2023), while the rest is taken from the Kaggle website (https://www.kaggle.com/datasets/borismarjanovic/price-volume-data-for-all-us-stocks-etfs, accessed on 7 September 2023).

For each time series, missing values are filled by propagating the previous observation forward to the next valid one (this data processing step is applied only to real-world time series). Subsequently, all the series are normalized to ensure that each value falls within the range from 0 to 1. This normalization is necessary to facilitate the comparison of forecast errors on an absolute scale. Finally, the dataset is split into training (800 observations) and test (200 observations) samples, starting from the last observation of each series in reverse order.

Not all the forecast models used in this study can directly accept a nontransformed training sample as input. Therefore, the time series are transformed to be represented as a supervised learning problem, allowing them to be compatible with the models used.

In Figure 4, the formation of input vectors from the processed data is illustrated. The input vector is constructed as follows: $X_{i}=\left(y_{1},\ldots ,y_{i+n-1}\right)$, where $y_{i}$ represents an observation from the time series, *n* is the number of steps considered for learning, and *i* is the index of the time series: i∈{1,…,N−n−1}. The desired output for the prediction model corresponds to the value of the next observation for each vector $X_{i}$: $Y_{i}=\left(y_{i+n}\right)$.

#### 3.2.2. Experiments with Artificial Time Series

The transformed data are divided into train and test periods based on the selected threshold (80% of the data for training and 20% for testing). As mentioned earlier, five models are employed to generate forecasts (ARIMA, LSTM network, Random Forest, XGBoost, and CNN). Using the forecasts generated by these models, the measures of realized predictability are computed. Additionally, based on the training dataset, measures of intrinsic predictability are calculated. Consequently, a dataset is compiled for each of the five models, containing values of predictability measures (five measures of realized predictability and six measures of intrinsic predictability) for each time series.

In Figure 5, an example of scatter plots is presented, depicting the relationship between measures of intrinsic and realized predictability for different forecasting models. The points on these plots are categorized into three classes based on the level of realized predictability, defined by the 0.33 quantile and 0.66 quantile (ensuring an equal number of points in each class). The points are color-coded according to their predictability class: red signifies a class with low predictability, where high forecast error values (y-axis) correspond to high values of the intrinsic predictability measure (x-axis). Conversely, green points represent a group of series characterized by low forecast errors and low values of the intrinsic predictability measure (indicating highly predictable series).

In Figure 5a, a scatter plot for the ARIMA forecasting model is presented. MAE is utilized as a measure of realized predictability, while permutation entropy (PE) is employed as a measure of intrinsic predictability. Notably, there exists a strong relationship between these predictability measures (Pearson correlation coefficient: 0.87, Spearman correlation coefficient: 0.97). Figure 5b–d illustrate analogous plots for the other models and their respective predictability measures.

Subsequently, all the datasets are merged, encompassing all forecasting models and predictability measures. Pearson pairwise correlation coefficients are then computed, and a correlation matrix is compiled (Figure 6). The obtained correlation coefficients are subjected to a test for statistical significance. This test aims to evaluate the null hypothesis $H_{0}:r_{xy}=0$, where $r_{xy}$ represents the pairwise correlation coefficient between variables *x* and *y*. The *p*-value is employed for this purpose. Essentially, the *p*-value signifies the likelihood of committing an error in rejecting the null hypothesis (Type I error). The chosen significance level, denoted as α, is set at 0.05.

Upon analyzing the correlation matrix in Figure 6, it becomes apparent that correlations exist between measures of realized predictability (MAE, MSE, RMSE) and measures of intrinsic predictability (PE, SE, SVD entropy, Approximate entropy, Sample entropy). In several instances, the correlation surpasses 0.8. Given that the *p*-values for these pairs are all below α=0.05, we are justified in rejecting the null hypothesis ($H_{0}$) at the 5% significance level. Consequently, a strong statistical relationship is established for numerous pairs of realized and intrinsic predictability measures.

The Pearson correlation test is parametric, requiring the assumption of normal distribution for each of the compared variables. When assessing correlations among variables with nonnormal distributions, including those measured on an ordinal scale, the Spearman rank correlation coefficient is more appropriate. Figure 7 illustrates the correlation matrix using the Spearman correlation coefficient. It is evident that a strong correlation exists between measures of realized and intrinsic predictability. The highest correlation value reaches 0.88 (in cases involving pairs such as SVD entropy and MAE, MSE, RMSE). In all three pairs, the *p*-values are considerably below 0.05, indicating that we can reject the null hypothesis at the 5% significance level.

The plots for several pairs of measures are displayed in Figure 8. As one can see, in the majority of cases, the highly predictable and medium predictable classes are distinctly distinguished from each other, while points from the low predictable class exhibit slight scattering.

Hence, in the case of the generated series, a strong correlation between measures of intrinsic and realized predictability is evident. The significance test outcomes affirm that the computed correlation coefficients for the mentioned pairs hold statistical significance.

#### 3.2.3. Experiments with Real-World Time Series

Similar experiments are also conducted on real-world time series. However, it is crucial to acknowledge a limitation that has been identified at this stage of the study.

Among the forecasting models used for the experiment, two models based on random forests are included: Random Forest and eXtreme Gradient Boosting. The nonlinear nature of random forests can provide an advantage over other algorithms, which is why these models are frequently employed for forecasting tasks. Nevertheless, it is essential to be aware that random forests are not capable of extrapolation. They can solely provide predictions based on the average of labels previously observed. This behavior can present challenges when the range of inputs during training and testing periods differs.

Considering this limitation, models based on random forests are excluded from the experiment involving real-world time series. This decision is made because the range of values in the test sample significantly diverges from the range of values in the training sample. As a result, analyzing forecast errors (measures of realized predictability) under such circumstances would be incorrect.

Figure 9 shows the correlation matrix for the remaining three models. It is evident that the correlation values have undergone substantial changes. However, for a few pairs (such as MAPE-SVD entropy), a significant correlation still exists. A statistical significance assessment of the correlation coefficient reveals that this correlation is significant at a 5% significance level. Moreover, several pairs exhibit a moderate relationship: MAE–Entropy, RMSE–Entropy, MAE–Permutation entropy, RMSE–Permutation entropy, MAPE–Permutation entropy, MAE–Approximate entropy, RMSE–Approximate entropy, MAE–Sample entropy, RMSE–Sample entropy, and MAPE–Sample entropy. For these pairs, the null hypothesis of zero correlation is also rejected at a 5% significance level.

In the correlation matrix with Spearman correlation coefficients (Figure 10), several pairs with moderate correlations can also be observed. These pairs include: MAE–Entropy, RMSE–Entropy, MAE–Permutation entropy, RMSE–Permutation entropy, MAPE–Permutation entropy, MSE–SVD entropy, MAE–Approximate entropy, RMSE–Approximate entropy, MAE–Sample entropy, and RMSE–Sample entropy. Additionally, the correlation between MAPE and SVD entropy shows a noticeable relationship. Importantly, all these correlations are significant at a 5% significance level.

It is worth noting that in some cases, the pair correlation for certain models (as shown in Figure 11 for the ARIMA model) reaches a value of 0.8 (Figure 11a) according to the Pearson correlation coefficient (for the MAE–Permutation entropy pair) and 0.74 (Figure 11b) according to the Spearman correlation coefficient (for the MAPE–SVD entropy and MAE–Approximate entropy pairs). The scatter plot for the MAPE–SVD entropy pair is depicted in Figure 12. Within this plot, the class of points with low predictability (red colored points) appears to be more dispersed throughout the space, while the other two classes reflect the relationship.

### 3.3. Experiments Summary

Experiments conducted on both generated and real-world time series reveal a positive relationship between measures of intrinsic predictability and measures of realized predictability. Specifically, a strong relationship is observed for multiple pairs (such as SVD entropy and MAE, MSE, RMSE, etc.) in the case of generated time series (Figure 7), and a noticeable relationship (in the case of MAPE–SVD entropy) is identified for real-world time series (Figure 10).

Hence, the connection between forecast errors (realized predictability) and the measures of intrinsic predictability is investigated. The conclusion about the presence of correlation proves highly beneficial in scenarios where evaluating time series complexity and its potential to be predictable with high level of accuracy is crucial. Estimating predictability using intrinsic predictability measures prior to the forecasting moment could notably streamline the process, minimizing time spent on model selection, parameter tuning, and other stages, especially for time series with evidently low predictability.

## 4. Conclusions

Motivated by the multitude of existing measures for estimating the predictability of different objects (time series, network links), this paper proposes an overview of existing works in this field. Specifically, we investigate predictability from two distinct perspectives: the *intrinsic* predictability, which represents a data property independent of the chosen forecasting model and serves as the highest achievable forecasting quality level, and the *realized* predictability, which is a selected quality metric for a specific data–model combination. We regard existing in the literature measures of intrinsic and realized predictability, applicable to univariate, multivariate, and categorical time series, as well as network link predictability estimation.

In the second part of our paper, we provide an analysis of the correlation between measures of realized and intrinsic predictability, which employs five forecasting models, five measures of realized predictability (forecast errors), and six measures of intrinsic predictability. Furthermore, to ensure the reliability of our findings, the experiments are conducted using both generated and real-world time series data. The presence of a strong statistically significant relationship (at a significance level of 5%) between predictability measures is obvious in the generated series. This relationship is observed not only across specific forecast models (as demonstrated in Figure 5), but also holds true for all models (Figure 8). The experiments involving real-world time series also confirm the presence of a relationship. Notably, the MAPE–SVD entropy pair exhibits a noticeable relationship (Figure 10), with the correlation coefficient being statistically significant at a significance level of 5%.

The limitations associated with using the regarded intrinsic predictability measures to estimate the complexity of time series before the forecasting moment are as follows. Firstly, all the regarded predictability measures are sensitive to data quality (missing values and measurement errors should be addressed during the preprocessing stage). Secondly, in the case of time series with changing over time nature, predictability measures should be applied carefully (with the possible extracting of time periods with different statistical properties).

The results of this research regarding the observed correlation are highly valuable for tasks that involve assessing the complexity of time series and their potential to be accurately predicted. Utilizing intrinsic predictability measures for predictability estimation prior to the actual forecasting moment can considerably reduce the time and effort required for tasks such as selecting an appropriate forecasting model and tuning parameters, especially for time series with evident low predictability. Further elaboration of this research is possible in the direction of inclusion of case studies illustrating how the obtained results can be applied to enhance the time series forecasting process. It will be also useful to expand the results of the experimental study on the network links predictability measures. This task is planned as a part of our future work.

## Figures and Tables

**Figure 1 entropy-25-01542-f001:**
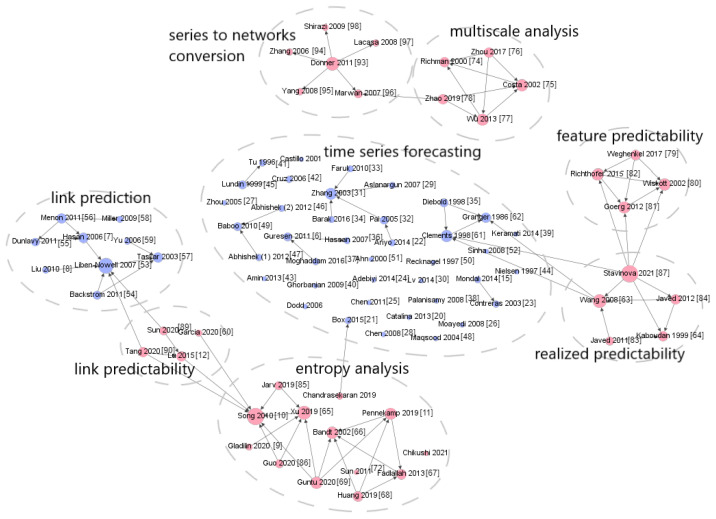
The citation network consisting of papers dedicated to forecasting and predictability assessment methods for different objects (time series, network links).

**Figure 2 entropy-25-01542-f002:**
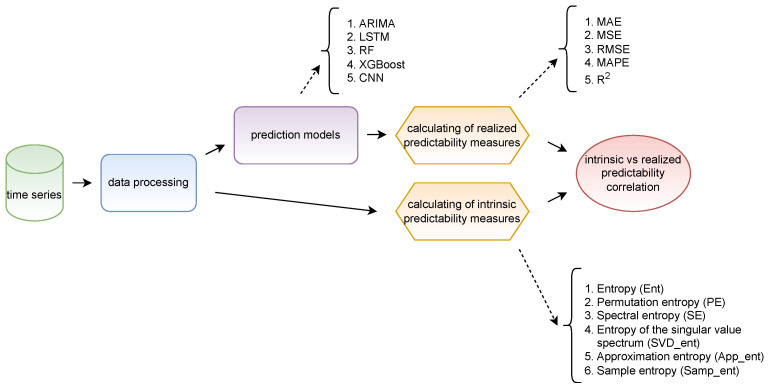
The pipeline of predictability correlation analysis.

**Figure 3 entropy-25-01542-f003:**
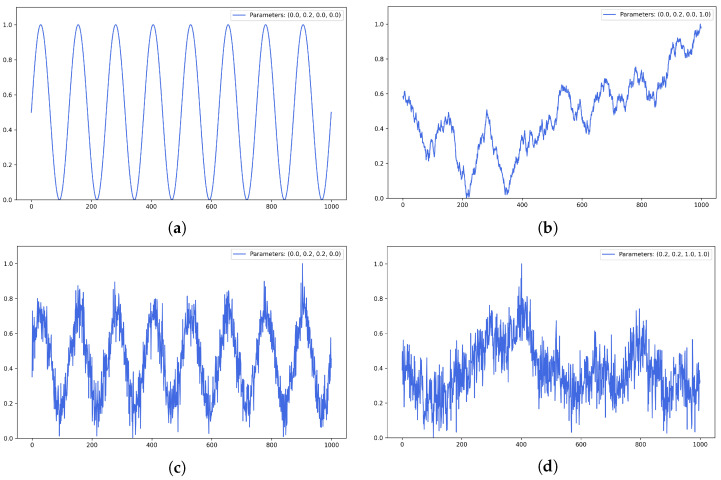
Examples of generated series: (**a**) only the periodic component, (**b**) the periodic component and a random walk, (**c**) the periodic component and randomness, (**d**) all the components.

**Figure 4 entropy-25-01542-f004:**
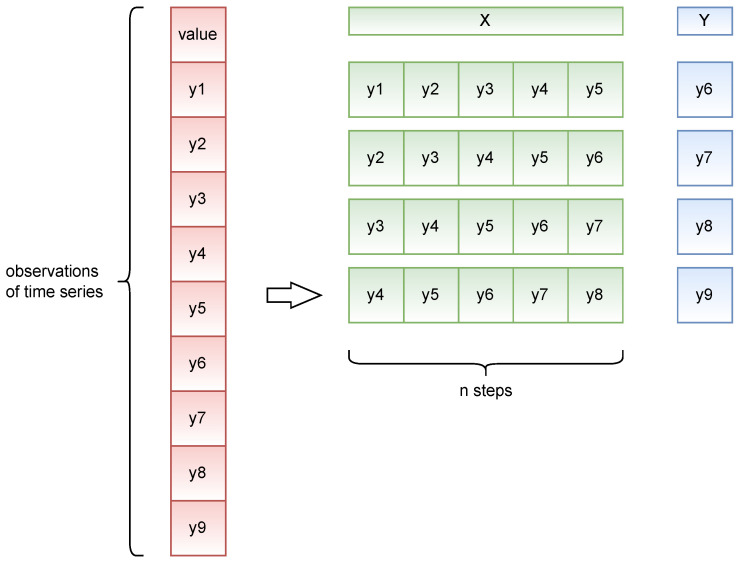
Example of transforming time series into input vectors with n=5.

**Figure 5 entropy-25-01542-f005:**
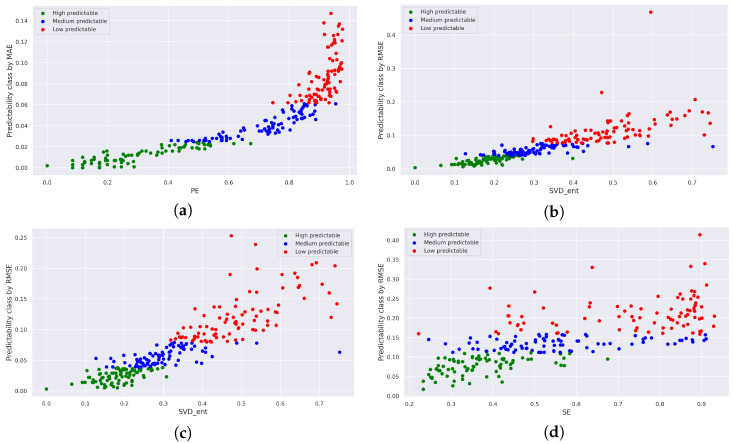
Scatter plots obtained by the forecasting models: (**a**) ARIMA forecast, with measures of MAE and PE, (**b**) LSTM forecast, with measures of RMSE and SVD entropy, (**c**) CNN forecast, with measures of RMSE and SVD entropy, (**d**) XGBoost forecast, with measures of RMSE and SE.

**Figure 6 entropy-25-01542-f006:**
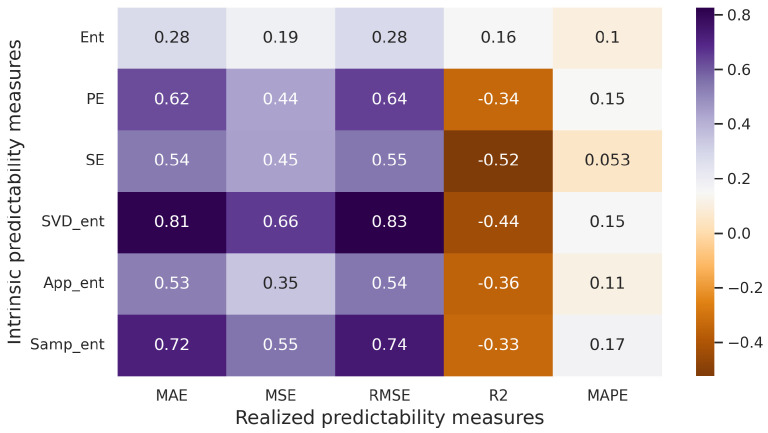
Correlation matrix (artificial time series, Pearson correlation coefficient).

**Figure 7 entropy-25-01542-f007:**
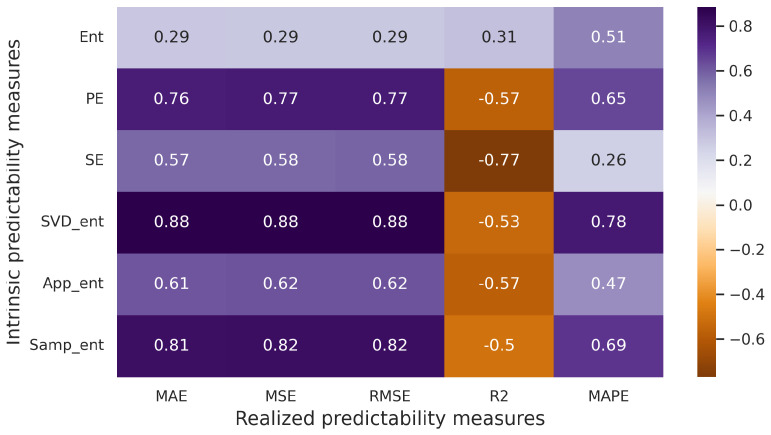
Correlation matrix (artificial time series, Spearman correlation coefficient).

**Figure 8 entropy-25-01542-f008:**
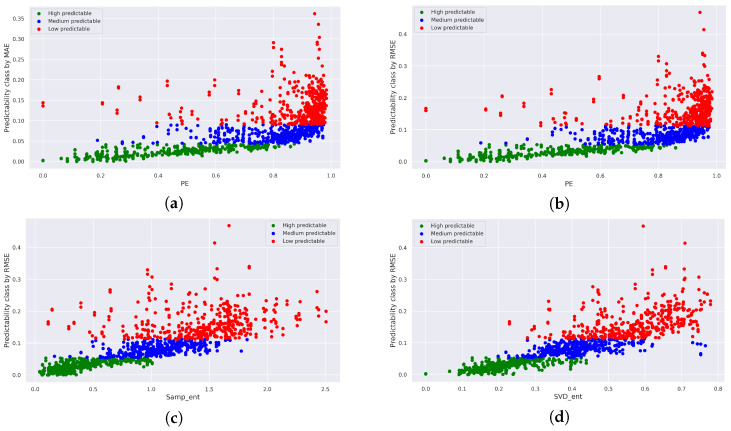
Scatter plots obtained after merging data from all forecasting models: (**a**) MAE and Sample entropy as measures, (**b**) RMSE and PE as measures, (**c**) RMSE and Sample entropy as measures, (**d**) RMSE and SVD entropy as measures.

**Figure 9 entropy-25-01542-f009:**
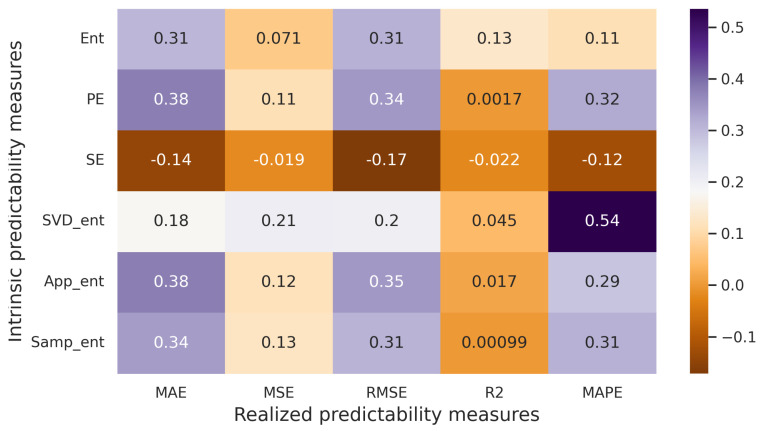
Correlation matrix (real-world time series, Pearson correlation coefficient).

**Figure 10 entropy-25-01542-f010:**
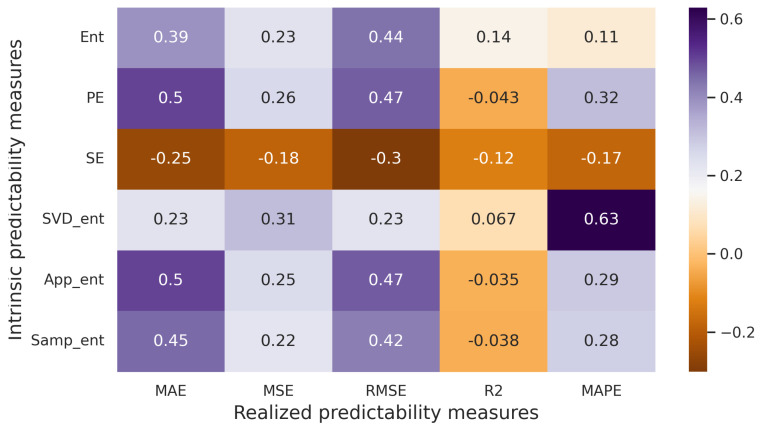
Correlation matrix (real-world time series, Spearman correlation coefficient).

**Figure 11 entropy-25-01542-f011:**
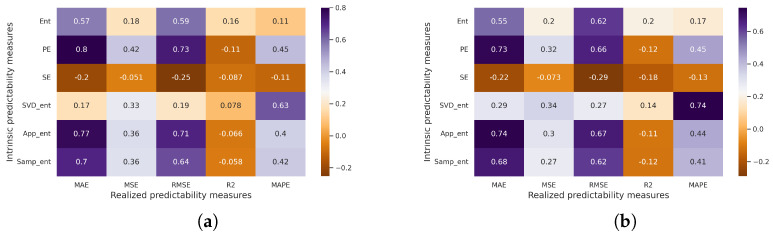
Correlation matrices for real-world time series (ARIMA model): (**a**) Pearson correlation coefficient, (**b**) Spearman correlation coefficient.

**Figure 12 entropy-25-01542-f012:**
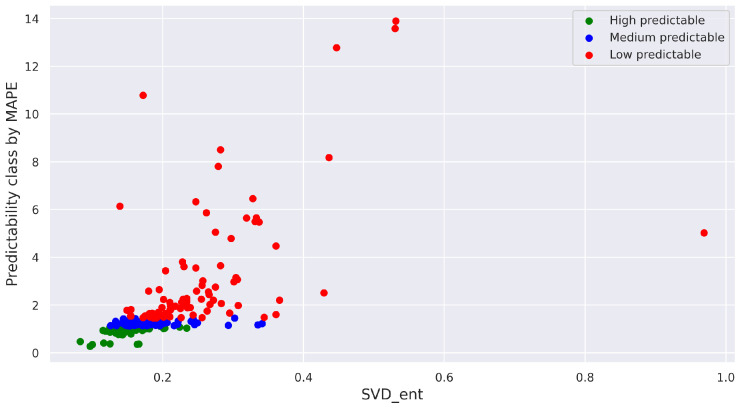
Scatter plot for the MAPE–SVD entropy pair obtained by ARIMA prediction model (real-world time series).

**Table 1 entropy-25-01542-t001:** Formulas for calculating forecast errors (measures of realized predictability).

Measure	Formula
Mean Absolute Error (MAE)	$\frac{1}{n}\sum_{i=1}^{n}|y\left(i\right)-\hat{y}\left(i\right)|$
Mean Squared Error (MSE)	$\frac{1}{n}\sum_{i=1}^{n}{\left(y\left(i\right)-\hat{y}\left(i\right)\right)}^{2}$
Root Mean Square Error (RMSE)	$\sqrt{\mathrm{MSE}}$
Mean Absolute Percentage Error (MAPE)	$\frac{1}{n}\sum_{i=1}^{n}\frac{|y\left(i\right)-\hat{y}\left(i\right)|}{y(i)}\cdot 100$
Coefficient of determination ($R^{2}$)	$\frac{\sum_{i=1}^{n}{\left({\hat{y}}_{i}-{\bar{y}}_{i}\right)}^{2}}{\sum_{i=1}^{n}{\left(y_{i}-\bar{y}\right)}^{2}}$

## Data Availability

The code for the artificial time series generation is available at https://github.com/Anthony-Cov/Timesergraph/blob/main/ArtSerGenerator.py, accessed on 7 September 2023. Part of the real-world time series is available at https://github.com/Anthony-Cov/Timesergraph/tree/main/RealWeekly, accessed on 7 September 2023, while the rest is taken from https://www.kaggle.com/datasets/borismarjanovic/price-volume-data-for-all-us-stocks-etfs, accessed on 7 September 2023.

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
