# Peer review of "Enhancing Predictability Assessment: An Overview and Analysis of Predictability Measures for Time Series and Network Links"

_entropy, 2023, doi:10.3390/e25111542_

Round 1
Reviewer 1 Report
Comments and Suggestions for Authors
The manuscript presents interesting information and my opinion about its quality is positive. I have the following comments.
1. The title. Predicting the unpredictable... If something is unpredictable then there is no determinism in it. Thus, it can not be predicted. So , please, think about the title.
2. The manuscript has two parts: review part where selected methods for forecasting time series and network links are presented and a part where the correlation between forecasting methods is calculated by use of generated and real-world time series. It is somehow a half-review and a half research article. The situation is not typical, but OK,a review can be written also in such a way.
3.The research area of time series forecasting and network links forecasting is large one and the selection of the methods for review reflects the preferred areas of the research of the authors. Despite this, the result is interesting and the manuscript could be published
4. The values of some of correlations are not very large. What is interesting here is the conclusion that estimating predictability using intrinsic predictability measures prior to the forecasting moment could
notably streamline the process, minimizing time spent on model selection, parameter tuning, and other stages, especially for time series with evidently low predictability. This is useful.
5. There are some print mistakes in the text and the literature is not made according the requirements of the journal (the DOI of the articles is missing). This has to be corrected.
6. Finally, my recommendation is minor revision before publication of the manuscript.
Reviewer 2 Report
Comments and Suggestions for Authors
The article "Unraveling Predictability Measures: A Comprehensive Analysis of Time Series and Network Links" provides a thorough and insightful overview of the various measures used to estimate predictability in different types of data, ranging from time series to network links. The paper distinguishes between intrinsic predictability, which is independent of the forecasting model chosen, and realized predictability, which represents a specific quality metric for a given pair of data and model. This dual perspective offers a comprehensive understanding of predictability assessment.
One of the notable strengths of this article is its extensive coverage of predictability measures across diverse data types, including univariate, multivariate, and categorical time series, as well as network links. This broad scope ensures that gain a holistic view of how these measures apply to various scenarios.
The experimental validation presented in the paper is another commendable aspect. The positive relationship observed between intrinsic and realized predictability in both generated and real-world time series data adds a valuable empirical dimension to the theoretical framework. The statistically significant correlation coefficients provide robust evidence for the validity of the proposed measures.
The conclusions drawn from the study highlight the practical implications of the research findings. The identification of strong relationships between specific predictability measures, such as SVD entropy and various forecast error metrics, offers concrete guidance for practitioners in assessing time series complexity. The recommendation to utilize intrinsic predictability measures before the forecasting phase is particularly noteworthy, as it has the potential to streamline the modeling process and save valuable time, especially for cases with low predictability.
One possible area for improvement could be the inclusion of real-world examples or case studies to illustrate how the proposed methodology can be applied in practical settings. Additionally, a discussion of potential limitations or challenges associated with using these predictability measures in specific contexts would further enhance the article's applicability.
In terms of expanding the title, a more specific title that reflects the key findings and contributions of the study could be considered. For example, "Enhancing Time Series Predictability Assessment: Insights from Intrinsic and Realized Measures" could encapsulate the essence of the research while also indicating its relevance to forecasting applications.
Overall, "Unraveling Predictability Measures" is a well-structured and informative article that advances understanding of predictability assessment in time series and network links. Its rigorous methodology and practical recommendations make it a valuable resource for researchers and practitioners in the field of time series analysis and forecasting.
